# Scaling of subcellular actin structures with cell length through decelerated growth

**Shane G McInally[1,2], Jane Kondev[2]\*, Bruce L Goode[1]\***

[1]Department of Biology, Brandeis University, Waltham, United States; [2]Department of Physics, Brandeis University, Waltham, United States

**Abstract** How cells tune the size of their subcellular parts to scale with cell size is a fundamental question in cell biology. Until now, most studies on the size control of organelles and other subcellular structures have focused on scaling relationships with cell volume, which can be explained by limiting pool mechanisms. Here, we uncover a distinct scaling relationship with cell length rather than volume, revealed by mathematical modeling and quantitative imaging of yeast actin cables. The extension rate of cables decelerates as they approach the rear of the cell, until cable length matches cell length. Further, the deceleration rate scales with cell length. These observations are quantitatively explained by a 'balance-point' model, which stands in contrast to limiting pool mechanisms, and describes a distinct mode of self-assembly that senses the linear dimensions of the cell.

## Introduction

The size of a cell's internal parts are scaled to its overall size. This size-scaling behavior has been demonstrated for organelles as well as large macromolecular assemblies, illustrating the broad importance of adapting the size of internal structures to the geometric dimensions of the cell (*Rafelski et al., 2012*; *Levy and Heald, 2010*; *Hazel et al., 2013*; *Good et al., 2013*; *Weber and Brangwynne, 2015*; *Greenan et al., 2010*; *Jorgensen et al., 2007*; *Decker et al., 2011*; *Neumann and Nurse, 2007*; *Lacroix et al., 2018*). A popular model of cellular scaling is the limiting pool mechanism, wherein maintaining a constant concentration of molecular components enables the subcellular structure to increase in size proportionally with cell volume (*Goehring and Hyman, 2012*; *de Godoy et al., 2008*) This allows larger cells to assemble larger structures, since the total number of molecular building blocks increases proportionally with cell volume. Additionally, this mechanism is biochemically simple because it does not require active processes that dynamically tune concentrations or activity levels of proteins involved in the construction . Indeed, cells appear to use a limiting pool mechanism to scale the size of their nucleoli, centrosomes, and mitotic spindles (*Hazel et al., 2013*; *Good et al., 2013*; *Weber and Brangwynne, 2015*; *Greenan et al., 2010*; *Decker et al., 2011*; *Lacroix et al., 2018*). However, limiting pool models cannot explain how the size of a linear subcellular structure scales with the linear dimensions of a cell, rather than its volume. Namely, these mechanisms predict that a two fold increase in the radius of a spherical cell will increase the length of a linear structure eight fold, following the eight fold increase in cell volume. This suggests that other mechanisms must account for how some subcellular structures are scaled with the linear dimensions of a cell.

Polarized actin cables in *S. cerevisiae* are an example of a linear structure that appear to grow to match the linear dimensions of the cell in order to effectively deliver secretory vesicles (*Moseley and Goode, 2006*). These cables are linear bundles of crosslinked actin filaments assembled by formins, which extend along the cell cortex and serve as tracks for intracellular transport of cargo from the mother cell to the growing bud, or daughter cell. Complementary sets of cables are assembled by two formins, Bni1 at the bud tip and Bnr1 at the bud neck (*Pruyne et al., 2004*). Throughout the cell

**\*For correspondence:**
kondev@brandeis.edu (JK);
goode@brandeis.edu (BLG)

**Competing interests:** The authors declare that no competing interests exist.

cycle, cables are continuously polymerized, turn over at high rates, and appear to grow until they reach the back of the mother cell (*Yu et al., 2011*; *Yang and Pon, 2002*; *Eskin et al., 2016*). This prompted us to more rigorously investigate the relationship between cable length and cell size.

We started by comparing cable lengths to the lengths of the mother cells in which they grew. Cables were imaged in fixed wild-type haploid cells using super-resolution microscopy. Cable lengths were measured from their site of assembly (the bud neck) to their distal tip in the mother cell (note that mother cell and cell are synonymous and used interchangeably from this point on) (*Figure 1—figure supplement 1A*). Average cable length and average cell length were remarkably similar (4.5 ± 0.3 μm and 4.5 ± 0.2 μm, respectively), suggesting a scaling relationship. However, we note that there was a wider range in cable lengths (2.0–8.7 μm) compared to cell lengths (3.7–5.5 μm), presumably because cables in fixed cells are at different stages of growth. Further, because cables grow along the cortex of an ellipsoid shaped cell, their length can exceed the length of the cell while not growing past the back of the cell. Therefore, a cable that grows from the bud neck to the back of the cell is expected to be slightly longer than the direct distance between these two points.

The observations above led us to ask whether the relationship between cable length and cell length is maintained as cell size increases. To address this, we compared cable lengths in haploid and diploid cells, and *cdc28-13$^{ts}$* temperature-sensitive mutants that grow abnormally large. Diploid mother cells had an ~2-fold increase in volume compared to haploid mother cells (81.8 ± 6.3 μm$^3$ and 44.9 ± 4.7 μm$^3$, respectively) (*Figure 1A,B and E*), consistent with previous studies (*Jorgensen et al., 2002*). The *cdc28-13$^{ts}$* strain exhibited a normal haploid mother cell size at the permissive temperature. However, this strain displayed a ~ 5-fold increase in volume (198.3 ± 5.5 μm$^3$ versus 40.9 ± 2.3 μm$^3$) after growth at the restrictive temperature (37°C) for 8 hr, followed by 1 hr of growth at the permissive temperature (25°C) to allow cell polarization and bud growth (*Figure 1C,D and E*; and *Figure 1—figure supplement 1B and C*; *Allard et al., 2018*). Accordingly, cell length increased with cell volume (*Figure 1F*). Cable length was greater in diploids (6.3 ± 0.7 μm) compared to haploids (4.5 ± 0.3 μm), and greater in induced (8.2 ± 0.4 μm) compared to unin-duced (4.3 ± 0.1 μm) *cdc28-13$^{ts}$* cells (*Figure 1G*). However, the distribution of cable lengths for all strains collapsed when we divided the lengths of cables by the lengths of the cells in which they grew (*Figure 1H and I*). These results strongly suggest that cables grow to a length that matches cell length.

Next, we used a power law analysis to rigorously test the scaling relationships of cable length with cell length and volume (*Figure 1J and K*). Generally, scaling relations can be described by the power law $y = Ax^a$, where $a$ is the scaling exponent that reflects the relationship between the two measured quantities, $x$ and $y$ (*Reber and Goehring, 2015*). This analysis revealed isometric scaling ($a_L = 0.91 \pm 0.03, R^2 = 0.50$) between cable length and cell length (*Figure 1J*), whereas scaling between cable length and cell volume was hypoallometric ($a_V = 0.36 \pm 0.01, R^2 = 0.46$) (*Figure 1K*).

To uncouple cell length from cell volume, we compared the length of cables in cells of different morphology. We computed the aspect ratio (the ratio of cell length to cell width) for the same cells analyzed above. This revealed that while some cells had nearly spherical morphologies, others had highly elongated morphologies (*Figure 1—figure supplement 2A and B*). Despite these differences in cell shape, the ratio of cable length to cell length, and the scaling exponents were similar for all cells (*Figure 1—figure supplement 2C–G*). Therefore, in cells of vastly different size and shape, the cable length directly scales with cell length, rather than with other dimensions such as cell surface area or volume.

We considered two distinct models to explain the control of cable length. In both models, the length of a cable is determined by competing rates of actin assembly ($k_+$) at the barbed ends of cables and disassembly ($k_-$) at the pointed ends of cables (*Figure 2A and B*). Therefore, at any given time, the extension rate of a cable is determined by the difference in its assembly and disas-sembly rates (*Figure 2B*). In the boundary-sensing model, the assembly rate is greater than the dis-assembly rate until the extending cable physically encounters the rear of the cell, causing one or both rates to abruptly change (*Figure 2C*, and *Figure 2—figure supplement 1A*; *Reber and Goehr-ing, 2015*). This model predicts that the cable extension rate will be constant until the cable tip encounters the back of the cell. In contrast, the balance-point model requires that either the assem-bly rate, the disassembly rate, or both rates are length-dependent, and defines steady state cable length as the point at which these two rates are balanced (*Figure 2D*, and *Figure 2—figure*

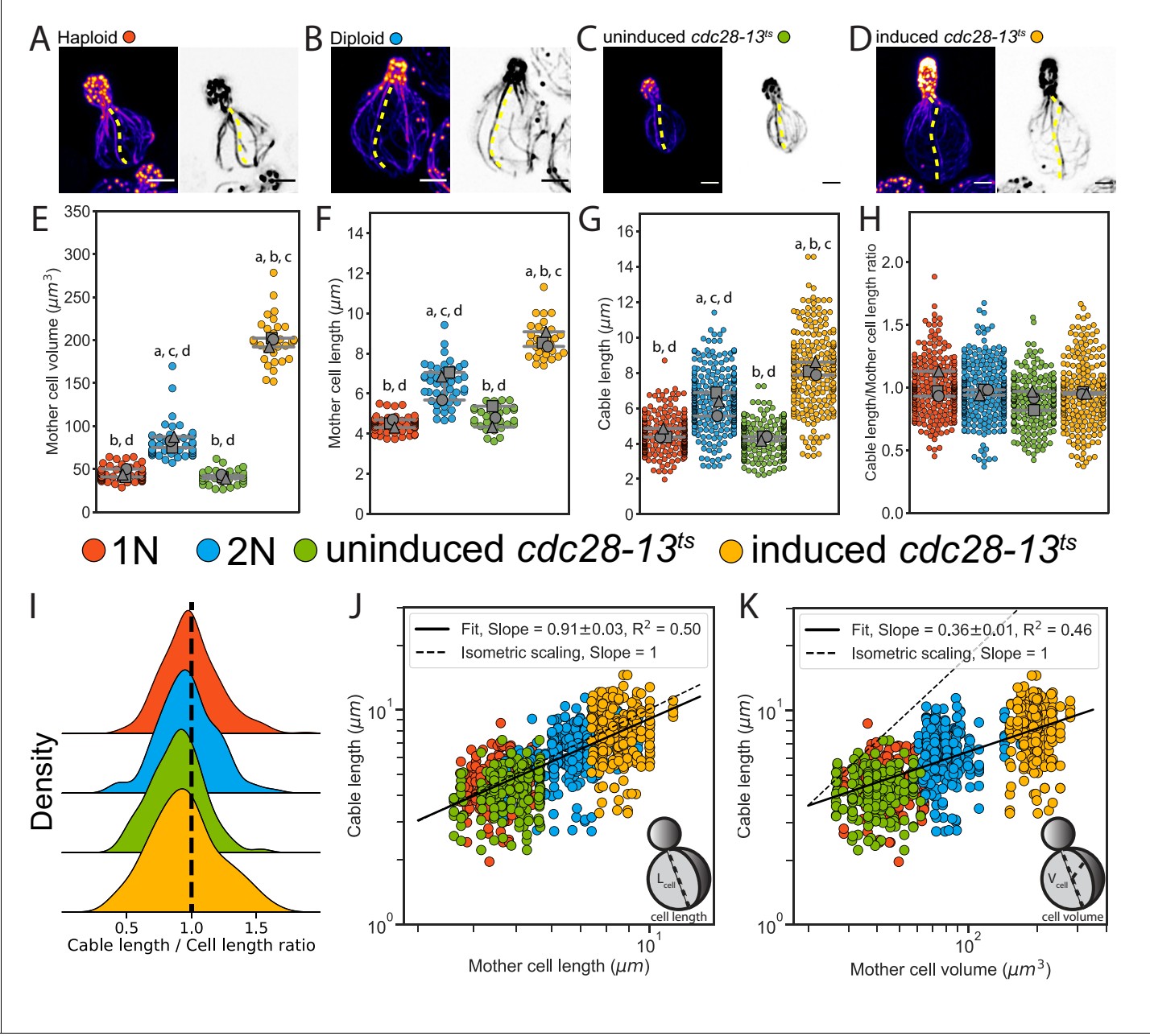

**Figure 1.** Actin cable length scales with cell length. (A–D) Representative images of haploid (A), diploid (B), uninduced *cdc28-13ts* (C), and induced *cdc28-13ts* (D) cells fixed and stained with labeled-phalloidin. Lengths of single actin cables are indicated (dashed lines) in maximum intensity projections (left, color) and single Z planes (right, inverted). Scale bar, 2 µm. (E–F) Mother cell volume (E) and length (F) measured in three independent experiments (≥30 cells/strain). Each data point is from an individual cell. Larger symbols represent the mean from each experiment. (G–H) Cable length (G) and ratio of cable length/cell length (H) measured from the same cells as in E and F (≥200 cables/strain). Each data point represents an individual cable. Larger symbols represent the mean from each experiment. Error bars, 95% confidence intervals. Statistical significance determined by students t-test. Significant differences (p≤0.05) indicated for comparisons with haploid ('a'), diploid ('b'), uninduced *cdc28-13ts* ('c'), and induced *cdc28-13ts* ('d'). Complete statistical results in *Figure 1—source data 1*. (I) Probability density functions for ratios in H. (J–K) Cable lengths plotted against mother cell length (J) or volume (K) on double-logarithmic plots and fit using the power-law. Hypothetical isometric scaling (dashed line) is compared to experimentally measured scaling exponent (solid line).

The online version of this article includes the following source data and figure supplement(s) for figure 1:

**Source data 1.** Complete results from statistical tests performed in this study.

**Figure supplement 1.** Changes to actin cable architecture in cells of different size.

**Figure supplement 2.** Changes to actin cable architecture in cells of different shape.

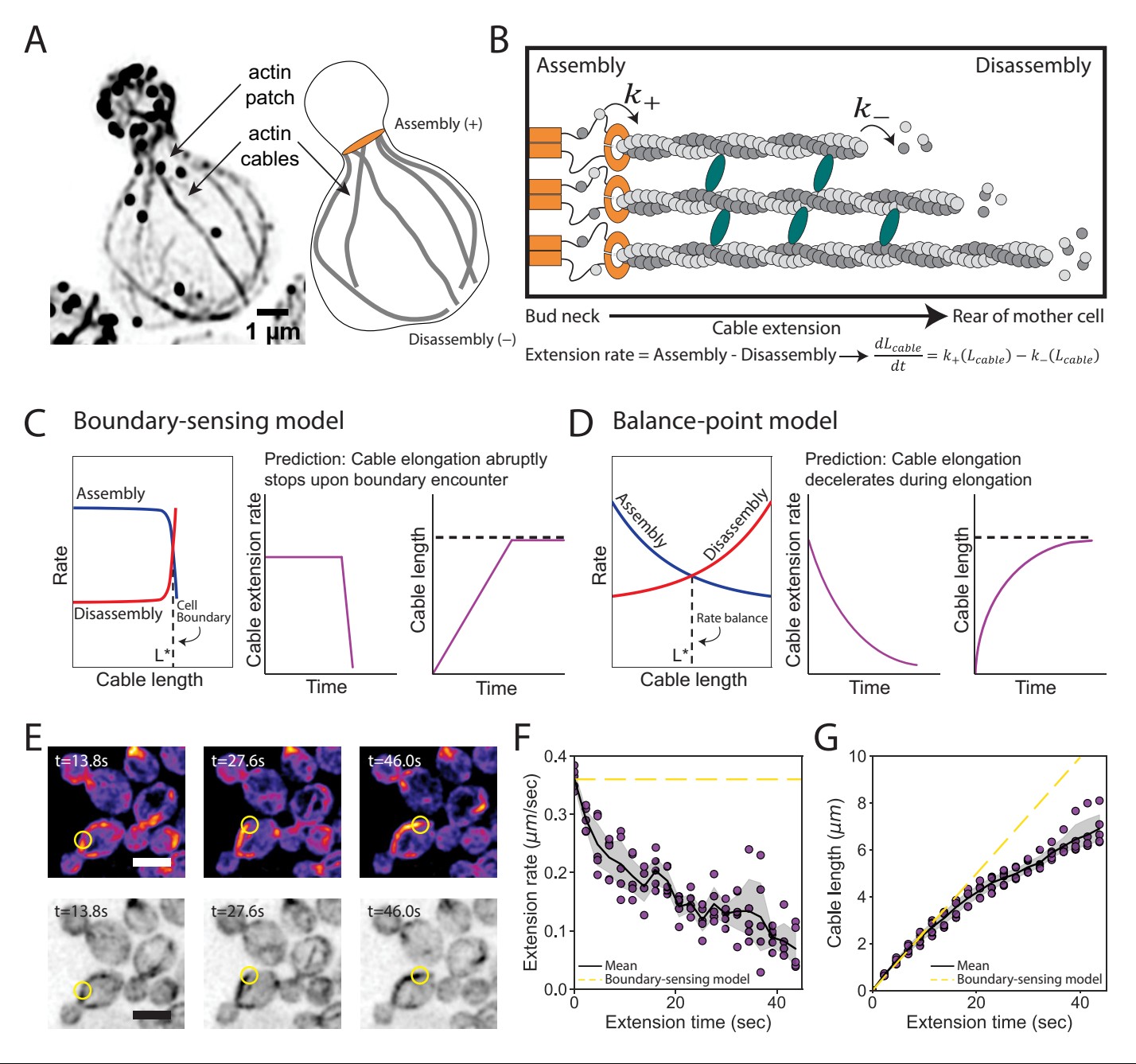

**Figure 2.** Models for control of actin cable length. (**A**) Actin staining in haploid cell (left) and cable traces (right). (**B**) Relevant parameters and equation for cable extension, where assembly ($k_+$) and disassembly ($k_-$) rates change as a function of cable length. Cables are polymerized by formins (orange) from actin monomers (gray), bundled by crosslinkers (blue), and disassembled by factors not shown. Cable extension rate is the difference in assembly and disassembly rates. (**C–D**) Two models for cable length control. Additional information in *Figure 2—figure supplement 1*. (**E**) Maximum intensity projection of haploid cells expressing cable marker (Abp140-GFP$^{Envy}$) shown in color (top panels) and inverted gray scale (bottom panels). Yellow circle highlights tip of elongating cable over time. Scale bar, 5 µm. (**F–G**) Extension rate (**F**) and length (**G**) measured in five independent experiments (n = 82 cables). Symbols at each time point represents mean for individual experiment. Solid lines and shading, mean and 95% confidence interval for all five experiments. Dashed yellow lines, predictions of boundary-sensing model in C.

The online version of this article includes the following figure supplement(s) for figure 2:

**Figure supplement 1.** Boundary-sensing and balance-point models of length regulation.

**Figure supplement 2.** 3D timeseries imaging workflow.

*supplement 1B*; *Mohapatra et al., 2016*). In clear contrast to the boundary-sensing model, this model predicts that the cable extension rate will steadily decrease as the cable lengthens.

To directly test the predictions of the two models, we used live imaging to track the tips of cables as they grew from the bud neck into the mother cell (*Figure 2E*, *Video 1*, and *Figure 2—figure supplement 2A–C*). Initially cables extended at 0.36 ± 0.02 µm s⁻¹, and as they grew longer their extension rates steadily decreased (*Figure 2F*, *Figure 2—figure supplement 2D*). Accordingly, we observed greater changes in cable length during earlier phases of cable growth (*Figure 2G*, *Figure 2—figure supplement 2E*). Thus, as cables lengthen their growth rate decelerates.

Note that we detected cables that were very short (<2 µm) by live imaging, which were not seen in our analysis of fixed cells. We expect that this is because shorter cables extend at a faster rate compared to longer cables and are therefore less prevalent in fixed cell populations.

Our experimental observations above support a balance-point model in which steady state cable length is reached when the assembly and disassembly rates are balanced. In this model, the rate of cable extension at any given time is given by the difference between the assembly and disassembly rates, which we call the feedback function, $f = k_+ - k_-$. To account for the observed scaling of cable length with cell length (*Figure 1H and I*), we assume that $f$ depends on the cable length ($L_{cable}$) scaled by the cell length ($L_{cell}$), that is $f(L_{cable}, L_{cell}) = f(L_{cable}/L_{cell})$. The steady state cable length $(L_{cable}^*)$ is reached when the feedback function equals zero, $f(L_{cable}^*/L_{cell}) = 0$. Therefore, the scale-invariant feedback function leads to the scaling of $L_{cable}^*$ with $L_{cell}$ seen in *Figure 1J*. (Further mathematical details in Materials and methods.)

Smy1 is a factor implicated in cable length control, and therefore we considered whether it might be required for cable deceleration. It has been reported that cables are longer in *smy1Δ* compared to wildtype cells, and that Smy1 directly inhibits Bnr1-mediated actin assembly (*Eskin et al., 2016*; *Chesarone-Cataldo et al., 2011*). Further, Smy1 is transported by myosin along cables to the bud neck where Bnr1 is anchored. Based on these observations, an 'antenna mechanism' has been proposed in which longer cables deliver more Smy1 to slow cable extension and limit cable length (*Mohapatra et al., 2015*). We confirmed the increase in cable length in *smy1Δ* cells (*Figure 3A*, and *Figure 3—figure supplement 1A and B*; *Eskin et al., 2016*), but found that cables continued to decelerate in the absence of Smy1 (*Figure 3B and C*). Furthermore, we observed an increase in the initial cable extension rate in *smy1Δ* (0.42 ± 0.04 µm s⁻¹) compared to wild-type cells (0.35 ± 0.02 µm s⁻¹) (*Figure 3D and E*). Interestingly, the initial extension rate in *smy1Δ* cells increased by the same magnitude (1.2-fold ± 0.2) as the measured increase in cable length (1.2-fold ± 0.1). Thus, Smy1 affects cable length by limiting the initial cable growth rate (*Figure 3F*) but does not provide the feedback that results in cable deceleration. Importantly, this does not rule out the possibility of other cellular factors acting through an antenna mechanism to control cable growth in a length-dependent manner.

Our model makes an interesting quantitative prediction for cables that have abnormally fast initial extension rates, such as those measured in *smy1Δ* cells above. Specifically, our model predicts that this increase in initial extension rate will lead to a proportional increase in the initial deceleration (see *Equation 7* in Materials and methods). Thus, the measured 1.2-fold increase in initial extension rate seen in *smy1Δ* cells is expected to lead to a 1.2-fold increase in initial deceleration of cables, shortly after they emerge from the bud neck. Indeed, linear fits to the cable extension rate, as a function of time over the first 10 s (i.e. the first few microns of cable extension), yield, $d_o^{smy1\Delta} = -0.018 \pm 0.010 \mu m/s^2$ and $d_o^{wt} = -0.015 \pm 0.005 \mu m/s^2$, for *smy1Δ* and wild-type cells, respectively (*Figure 3B and C*). The ratio of these two, $d_0^{smy1\Delta}/d_0^{wt} = 1.2 \pm 0.7$, matches the ratio of the initial extension rates, $f(0)^{smy1\Delta}/f(0)^{wt} = 1.2 \pm 0.2$. Therefore, these data lend additional quantitative support for our model.

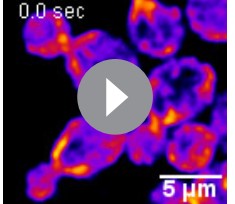

**Video 1.** Maximum intensity projection of haploid cells expressing a cable marker (Abp140-GFP^Envy) shown in color. Yellow circle highlights tip of an elongating cable over time. Video is played at 7 frames per second and time (seconds) is indicated in the top left corner. Scale bar, 5 µm.

https://elifesciences.org/articles/68424#video1

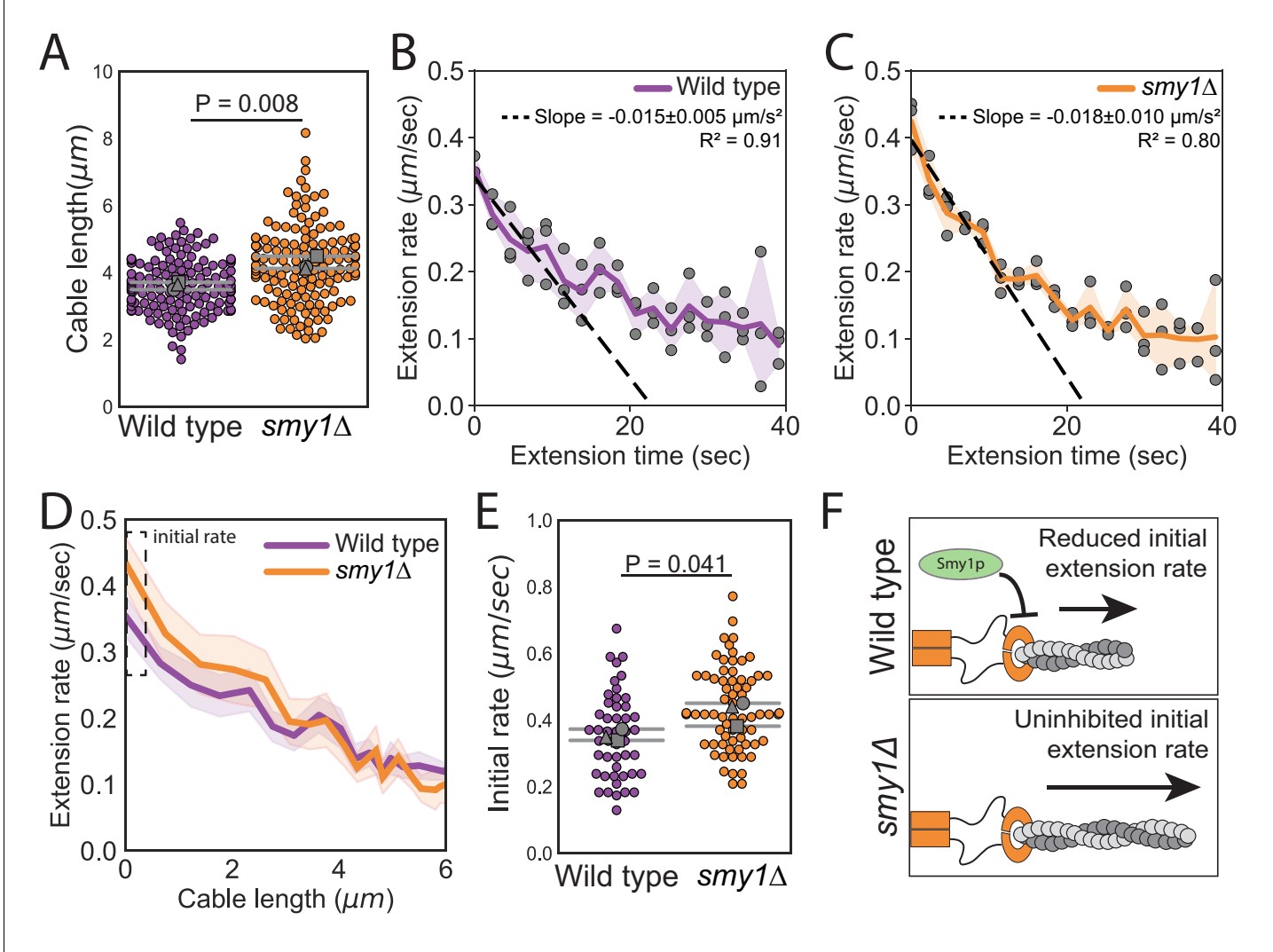

**Figure 3.** Smy1 controls initial cable extension rate. All data are from three independent experiments. (**A**) Cable lengths (≥130 cables/strain). Each data point represents an individual cable. Larger symbols, mean from each experiment. Error bars, 95% confidence intervals. Statistical significance determined by students t-test. (**B–C**) Cable extension rates for wildtype (**B**) and *smy1Δ* (**C**) yeast (≥47 cables/strain). Symbols, mean from each experiment. Solid lines and shading, mean and 95% confidence interval for all experiments. Deceleration rates were derived from the slopes (±95% CI) of the dashed lines, which were determined by linear regression using the first ~10 s of extension. (**D**) Average extension rate as a function of cable length. Solid lines and shading, mean and 95% confidence interval for all experiments. Dashed box highlights region of no overlap in confidence intervals. (**E**) Initial cable extension rate for each strain. Small symbols, individual cables. Larger symbols, mean from each experiment. Error bars, 95% confidence intervals. Statistical significance determined by students t-test. (**F**) Cartoon comparing cable extension in wildtype and *smy1Δ* cells. The online version of this article includes the following figure supplement(s) for figure 3:

**Figure supplement 1.** Altered actin cable length and architecture in *smy1Δ* cells.

A key prediction of our balance-point model is that cable extension rates should depend on cell length, that is a cable of a given length should grow faster (or slow down more gradually) in longer cells compared to shorter cells (*Figure 4A*, top). Further, it predicts that the cable extension rate profiles from cells of different lengths will collapse when cable length is normalized to cell length (*Figure 4A*, bottom; predictions of model derived in Materials and methods). To test these predictions, we compared cable extension dynamics in uninduced and induced *cdc28-13ts* cells (*Figure 4B and C*, *Figure 4—figure supplement 1A and B*, and *Videos 2*, *3*, *4*). When cables began to grow,

they extended at similar rates in shorter and longer cells (*Figure 4—figure supplement 1C*). However, as the cables grew longer, they decelerated more gradually in the longer cells (*Figure 4D–F*). This led to longer cables in longer cells (*Figure 4—figure supplement 1D*). Linear regression analysis revealed that there is a nearly 2-fold greater initial deceleration in the shorter, uninduced *cdc28-13^{ts}* cells ($d_0^{uninduced} = -0.019 \pm 0.005 \mu m/s^2$) compared to the longer, induced *cdc28-13^{ts}* cells ($d_0^{induced} = -0.010 \pm 0.003 \mu m/s^2$). To determine how deceleration changes with respect to cell length, we compared the ratio of initial deceleration and cell length in induced ($L_{induced} = 8.2 \pm 0.4 \mu$m) and uninduced ($L_{uninduced} = 4.3 \pm 0.1 \mu$m) *cdc28-13^{ts}* cells. We found that the initial deceleration rate is inversely proportional to cell length ($d_0^{uninduced}/d_0^{induced} = 2 \pm 1$; $(L_{uninduced}/L_{induced})^{-1} = 1.9 \pm 0.1$), consistent with the predictions of our balance-point model (*Figure 4D and E* and *Equation 7* in Materials and methods). Further, once cable length was normalized to cell length, cables extended with similar dynamics (*Figure 4G*), as predicted by our model.

Collectively, our observations demonstrate that cables grow until their length matches the length of the cell, and that this is achieved by length-dependent deceleration of cable extension. The precise mechanism providing the feedback to enable cable deceleration is not yet clear. One possibility is that it is controlled by a gradient of actin disassembly-promoting activity that is highest at the rear of the cell. Such a gradient could be established by the retrograde transport of disassembly factors on cables, leading to their release at the rear of the cell. This would produce a higher concentration of disassembly factors, and greater disassembly rate for cables, at the back of the cell. An alternative possibility is a reaction-diffusion mechanism, achieved by anchoring an activator of disassembly factors (such as a kinase) at the rear of the cell while having an inhibitor (such as a phosphatase) in the cytosol. This would be similar conceptually to how Ran GTPase gradients form around chromatin (*Kalab et al., 2002*), although it would require additional features to produce the scaling that we observe (*Ben-Zvi et al., 2011*). Either of these two mechanisms (retrograde transport or modified reaction-diffusion) has the potential to create a gradient that is shallower in longer cells compared to shorter cells, accounting for the cell-length-sensitive cable deceleration (*Figure 2—figure supplement 1D*). This mechanism also would allow cables to sense the rear of the cell without requiring physical interactions with that boundary. A third possibility, which is not mutually exclusive with either mechanism above, would be length-dependent inhibition of cable assembly, that is an antenna mechanism, albeit one that is dependent on cellular factors other than Smy1 (*Mohapatra et al., 2015*).

It has recently been shown for other subcellular structures (e.g. nucleus, spindle, centrosome, and nucleolus) that their sizes scale with cell volume, and this scaling is explained by limiting pool models (*Hazel et al., 2013*; *Good et al., 2013*; *Weber and Brangwynne, 2015*; *Decker et al., 2011*; *Neumann and Nurse, 2007*; *Lacroix et al., 2018*). However, we found that polarized actin cables scale with cell length rather than volume. This length control cannot be explained by a limiting pool mechanism, and instead is explained, both theoretically and experimentally, by a balance-point model. These results reveal a new strategy by which cells solve engineering challenges, enabling them to scale internal structures with the linear dimensions of the cell (*Kirschner et al., 2000*). Similar principles may underlie the length control of other polarized, linear actin structures, such as filopodia and stereocilia. Further, related strategies may be used to control the growth of radial microtubule arrays that reach the cell periphery (*Lacroix et al., 2016*; *Wühr et al., 2010*), and may explain the scaling relationships observed between flagellar length and cell length (*Bauer et al., 2021*) and between contractile ring diameter and cell diameter (*Kukhtevich et al., 2020*). Ultimately, the model of size control that we have presented here expands our understanding of the mechanisms used by cells to sense specific aspects of their geometry, including length, surface area, and volume, to assemble structures that scale with these different dimensions (*Rieckhoff et al., 2020*; *Brownlee and Heald, 2019*).

## Materials and methods

### Key resources table

*Continued*

| Reagent type (species) or resource | Designation | Source or reference | Identifiers | Additional information |
|---|---|---|---|---|
| Reagent type (species) or resource | Designation | Source or reference | Identifiers | Additional information |
| Strain, strain background (*S. cerevisiae*) | See: *Supplementary file 1* | This paper | NCBITaxon:4932 | Strains maintained in the Goode lab |
| Chemical compound, drug | Alexa Fluor 488- phalloidin | Life Technologies | A12379 | |
| Chemical compound, drug | Alexa Fluor 568-phalloidin | Life Technologies | A12380 | |
| Recombinant DNA reagent | pFA6a-link-GFPEnvy-SpHis5 | PMID:25612242 | RRID:Addgene_60782 | |
| Recombinant DNA reagent | pFA6a-TRP1 | PMID:9717241 | RRID:Addgene_41603 | |

## Plasmids and yeast strains

All strains (see *Supplementary file 1*) were constructed using standard methods. To integrate a bright GFP variant (GFP$^{Envy}$) at the C-terminus of the endogenous *ABP140* gene, primers were designed with complementarity to the 3' end of the GFP$^{Envy}$ cassette and the C-terminal coding region of *ABP140*. PCR was used to generate amplicons from the pFA6a-link-GFP$^{Envy}$-SpHis5 (*Slubowski et al., 2015*) template that allow for selection of transformants using media lacking histidine. The parent strains, BGY12 (haploid) and *cdc28-13$^{ts}$*, were transformed with PCR products, and transformants were selected by growth on synthetic media lacking histidine. Similarly, *smy1Δ* strains were generated by replacement of *SMY1* with the *TRP1* auxotrophic marker by designing primers with complementarity to regions 40 base-pairs immediately up-stream and down-stream of the *SMY1* coding region (*Longtine et al., 1998*). Deletion of *SMY1* was confirmed by genomic PCR with primers specific to the *TRP1* promoter and the 5'UTR region of *SMY1*. The *cdc28-13$^{ts}$* strain was a generous gift from Brian Graziano (UCSF). pFA6a-link-GFP$^{Envy}$-SpHis5 was a gift from Linda Huang (UMass Boston) (Addgene plasmid # 60782; http://n2t.net/addgene:60782; RRID:Addgene_60782).

## Induction of cell size changes

To induce enlargement of mother cells, *cdc28-13$^{ts}$* cells were grown at the permissive temperature (25°C) overnight in synthetic complete media (SCM), then 10µL of overnight culture was diluted into 5mL of fresh SCM. Cultures were then shifted to the restrictive temperature (37°C) for 8 hr (except for the experiments in *Figure 1—figure supplement 1B and C*, where cultures were also shifted for only 4 hr). After this induction, cells were returned to the permissive temperature (25°C) for 1 hr of growth to allow cell polarization and bud growth, and then fixed or mounted for live-cell imaging.

## Quantitative analysis of actin cable length and architecture in fixed cells

Strains were grown at 25°C to mid-log phase (OD$_{600}$ ~0.3) in yeast extract/peptone/dextrose (YEPD), or were first induced for cell size changes as indicated above. Then cells were fixed in 4.4% formaldehyde for 45 min, washed three times in phosphate-buffered saline (PBS), and stained with Alexa Fluor 488- phalloidin or Alexa Fluor 568-phalloidin (Life Technologies, Grand Island, NY) for ≥24 hr at 4°C. Next, cells were washed three times in PBS and imaged in mounting media (20 mM Tris, pH 8.0, 90% glycerol). 3D stacks were collected at 0.22 µm intervals on a Zeiss LSM 880 using Airyscan super-resolution imaging equipped with 63 × 1.4 Plan-Apochromat Oil objective lens. 3D stacks were acquired for the entire height of the cell. Airyscan image processing was performed using Zen Black software (Carl Zeiss). ImageJ was used to generate inverted greyscale and maximum projection images for analysis. Next ImageJ was used to manually trace each individual cable, from the bud neck to their terminus in the mother cell. The 3D stack was used to differentiate between cables that overlapped and to precisely determine both the origins and distal tips of the cables. For length

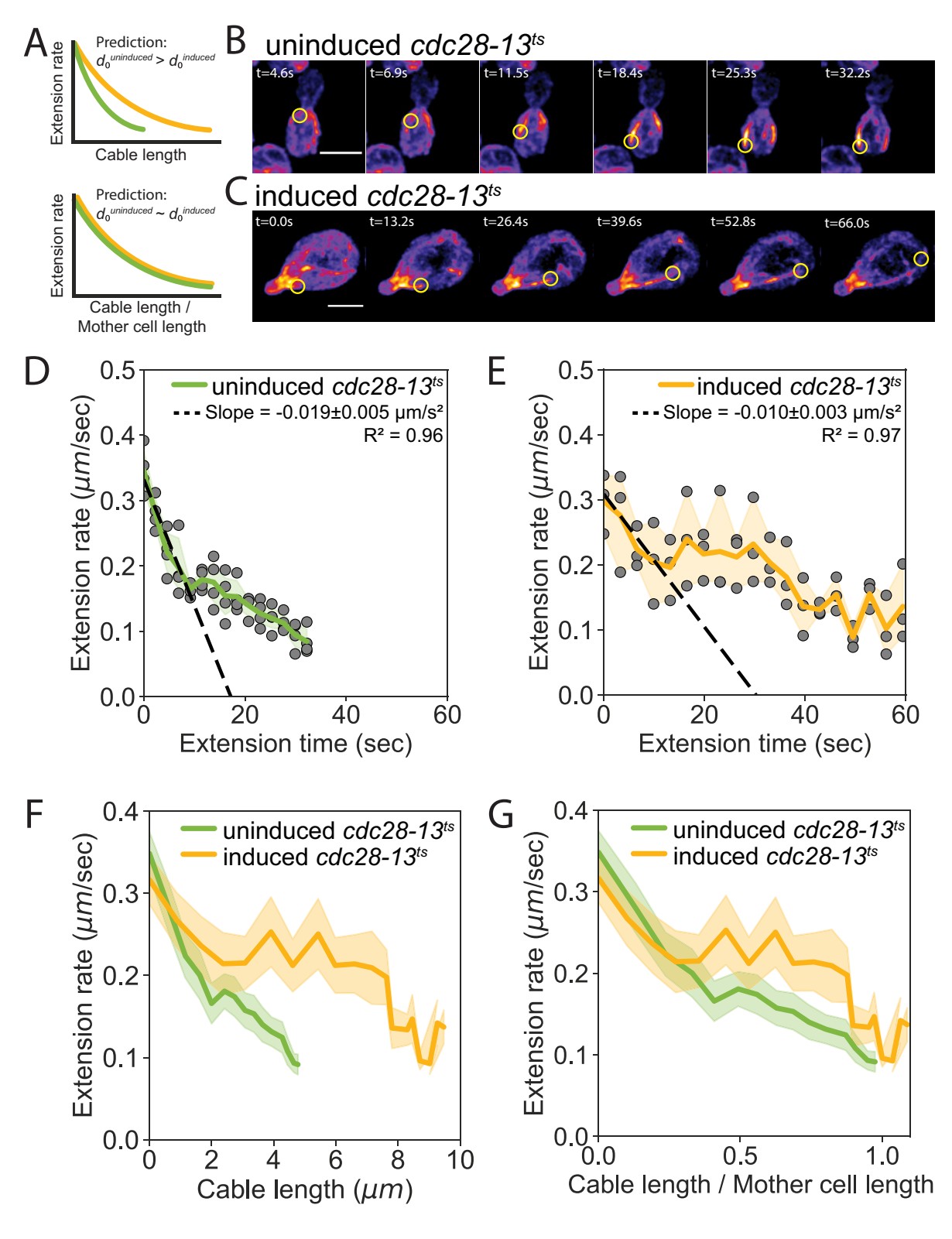

**Figure 4.** Cell length-dependent deceleration of actin cable growth. (A) Predictions of balance-point model comparing how cable deceleration ($d_0$) changes as a function of cable length (top graph) in shorter (green curve) and longer (yellow curve) cells. This difference in the deceleration profiles is eliminated when cable length is normalized to cell length (bottom graph). (B–C) Maximum intensity projections of uninduced (B) and induced *cdc28-13ts* (C) cells expressing cable marker (Abp140-GFP^Envy). Yellow circle highlights tip of elongating cable over time. Scale bar, 5 μm. (D–E) Cable

*Figure 4 continued on next page*

*Figure 4 continued*

extension rates for uninduced (D) and induced *cdc28-13^ts* (E) cells, from at least three independent experiments (≥57 cables/strain). Symbols and shading, mean and 95% confidence intervals for all experiments. Deceleration rates were derived from the slopes (±95% CI) of the dashed lines, which were determined by linear regression using the first ~10 s of extension. (F–G) Average extension rates in uninduced and induced *cdc28-13^ts* cells (data from experiments in D and E) plotted as a function of cable length (F), or the ratio of cable length/cell length (G). Solid lines and shading, mean and 95% confidence interval for all experiments.

The online version of this article includes the following figure supplement(s) for figure 4:

**Figure supplement 1.** Cable extension dynamics are cell-length dependent.

analysis, we included every discernable cable in the cell that extended from the bud neck to some endpoint in the mother cell; the only cables excluded were the minority that became closely intertwined with other cables making it impossible to resolve their individual lengths. Then the xy-coordinates for each cable trace were exported into custom written Python scripts to compute cable length. Cell length was determined by measuring the distance from the bud neck to the distal end of the mother cell. Cell width was determined by measuring the widest point perpendicular to the cell length axis. Cell height was determined from the number of slices in the 3D stack and the interval size between slices. These values were recorded and imported into custom Python scripts to compute the ratio of cable length to mother cell length, the cell volume (using the ellipsoid formula), the aspect ratio (cell length/cell width), and to fit the scaling exponent for cable length versus mother cell length, width, and volume. For cell shape analysis, cells were binned based on their aspect ratio rounded to the nearest quarter value.

## Live-cell imaging and quantitative analysis of actin cable extension rate

Strains were grown at 25°C to mid-log phase (OD$_{600}$ ~0.3) in either YEPD, or were first induced for cell size changes as indicated above, then harvested by centrifugation (30 s, 9000 x g). Media was decanted and cells were resuspended in 50 µL fresh media. Cells (~5 µL) were mounted onto 1.2% agarose pads made with SCM, and images were acquired on a Nikon i-E upright confocal microscope equipped with a CSU-W1 spinning-disk head (Yokogawa, Tokyo, Japan) and an Andor Ixon 897 Ultra CCD camera controlled by Nikon NIS-Elements Advanced Research software using a 100x, 1.45 NA objective. 3D stacks were acquired at 0.3 µm intervals for approximately half of the cell height with no time delay for 2 min (approximately 0.30–0.43 frames per second). Images were processed in ImageJ by generating maximum intensity projections of each stack and applying a Gaussian blur (sigma = 1) to facilitate manual tracking of cable tips. Cables included for analysis were those whose tips could be resolved in every frame, from when they emerged from the bud neck and until they stopped extending. Cables that could not be reliably tracked (e.g. dim cables, overlapping cables that prevented tracking of their tips, or cables that grew into regions not captured in the 3D stack) were excluded from the analysis. Individual cable trajectories were imported into custom Python scripts to compute the distance the cable tip travelled between each frame, the rate of extension between each frame, and the total distance travelled. The boundary-sensing model prediction depicted in *Figure 2F* was determined by

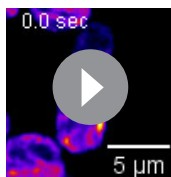

**Video 2.** Maximum intensity projections of uninduced *cdc28-13^ts* cell expressing cable marker (Abp140-GFP^Envy) shown in color (top panels). Yellow circle highlights tip of elongating cable over time. Video is played at 7 frames per second and time (s) is indicated in the top left corner. Scale bar, 5 µm.
https://elifesciences.org/articles/68424#video2

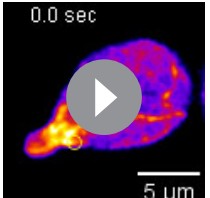

**Video 3.** Maximum intensity projections of induced *cdc28-13^ts* cell expressing cable marker (Abp140-GFP^Envy) shown in color (top panels). Yellow circle highlights tip of elongating cable over time. Video is played at 7 frames per second and time (s) is indicated in the top left corner. Scale bar, 5 µm.
https://elifesciences.org/articles/68424#video3

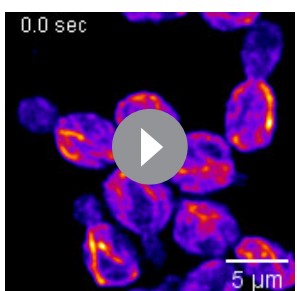

**Video 4.** Maximum intensity projections of uninduced *cdc28-13*<sup>ts</sup> cells expressing cable marker (Abp140-GFP<sup>Envy</sup>) shown in color (top panels). Yellow circle highlights tip of elongating cable over time. Video is played at 7 frames per second and time (s) is indicated in the top left corner. Scale bar, 5 μm.
https://elifesciences.org/articles/68424#video4

plotting the mean initial cable extension rate as a function of time. The boundary-sensing model prediction depicted in *Figure 2G* was determined by using linear regression to measure the slope from the first ~10 s of cable extension. Initial cable extension rates (*Figure 3C* and *Figure 4—figure supplement 1C*) were determined by computing the extension rate measured during the first time interval.

## Mathematics of the balance point model

The rate of change of the cable length with time is given by the difference between the assembly ($k_+$) and disassembly ($k_-$) rates,

$$\frac{dL_{cable}}{dt} = k_+(L_{cable}, L_{cell}) - k_-(L_{cable}, L_{cell}). \quad (1)$$

where we have made explicit the possibility that one or both rates depend on the length of the extending cable ($L_{cable}$) and the cell length ($L_{cell}$). The steady state length $L_{cable}^*$ is the cable length at which the assembly and disassembly rates are the same.

To account for the scaling of the steady state length with the cell length (as observed in *Figure 1H,I and J*), we make an additional assumption, namely that the feedback function, $f \equiv k_+ - k_-$, which determines the rate of cable extension, is a function of the ratio of the cable length to the cell length, that is $f(L_{cable}, L_{cell}) = f(L_{cable}/L_{cell})$. Thus, our mathematical model of cable length control is described by the differential equation:

$$\frac{dL_{cable}}{dt} = f(L_{cable}/L_{cell}), \quad (2)$$

which is graphically summarized in *Figure 2—figure supplement 1C*.

At the molecular scale, this feedback could be accomplished with a constant rate of cable assembly and a variable rate of disassembly controlled by a gradient of depolymerizing activity that is highest at the back of the cell; see *Figure 2—figure supplement 1C*. In this mechanism, as the cable lengthens its distal end is subject to increasingly stronger depolymerizing activity. Further, the profile or decay-length of the depolymerizing gradient needs to scale with cell length. Such scaling of a cellular gradient with the linear distance between the two poles of the cell has been observed for the protein Bicoid in different size embryos, from different species of flies (*Gregor et al., 2005*). Other experimental observations and theoretical models of such scale-invariant gradients are reviewed in *Ben-Zvi et al., 2011*.

*Figure 4F and G* are a direct test of our model. In *Figure 4F*, we observe that the cable extension rate is dependent on cell length, consistent with *Equation 1*. In *Figure 4G*, we see that the two feedback functions, from cells of different size, collapse to a single function when the cable lengths are scaled by the cell length.

The scaling property of the feedback function immediately leads to scaling of steady state cable length with cell length. Namely, in steady state, the right-hand side of *Equation 1* is zero, which implies $f\left(L_{cable}^*/L_{cell}\right) = 0$. If the zero of the feedback function is $x^*$ (i.e. $f(x^*) = 0$), then the steady state length $L_{cable}^* = x^* L_{cell}$, which is the scaling relation we observe in *Figure 1H and J* between the steady state length and the cell length.

The scaling property of the feedback function also makes a prediction for the initial rate of cable extension in cells of different size. Namely, for small cable lengths, when $L_{cable} \ll L_{cell}$, we can expand *Equation 2* into a Taylor series

$$\frac{dL_{cable}}{dt} = f\left(\frac{L_{cable}}{L_{cell}}\right) \approx f(0) + f'(0)\frac{L_{cable}}{L_{cell}}, \quad (3)$$

which states that the initial cable extension decreases linearly with the cable length (since $f'(0)$ is negative) and is inversely proportional to cell length, $L_{cell}$.

*Equation 3*, with the initial condition $L_{cable}(t = 0) = 0$, can be solved for the cable length as a function of time,

$$L_{cable}(t) = L_{cell}\frac{f(0)}{f'(0)}\left[e^{\frac{f'(0)}{L_{cell}}t} - 1\right],\tag{4}$$

which in turn yields, by differentiation, an exponentially decreasing in time extension rate:

$$\frac{dL_{cable}}{dt} = f(0)e^{\frac{f'(0)}{L_{cell}}t}.\tag{5}$$

Since *Equations 4 and 5* only hold at early times when the cable length is much smaller than the cell length (roughly, first 10 s of cable extension; see *Figure 2G*), we can further simplify *Equation 5* by expanding it into a Taylor series:

$$\frac{dL_{cable}}{dt} = f(0) + \frac{f(0)f'(0)}{L_{cell}}t.\tag{6}$$

*Equation 6* makes very specific predictions about the initial deceleration of cable extension, in particular our model (*Equation 2*) predicts that the initial deceleration

$$d_0 = \frac{d^2L_{cable}}{dt^2}\Big|_{t=0} = \frac{f(0)f'(0)}{L_{cell}}\tag{7}$$

scales inversely with the cell length, and proportionally with initial cable extension rate. Indeed, these predictions are supported in two independent experimental tests of this model. Our analysis of *smy1Δ* cells indicates that increasing $f(0)$, while $f'(0)$ and $L_{cell}$ are fixed, leads to a proportional increase in initial deceleration rate. Additionally, our analysis of induced and uninduced *cdc28-13^{ts}* cells, where $L_{cell}$ increases ~2-fold, while $f(0)$ and $f'(0)$ are fixed, leads to a two fold difference in initial deceleration.

Finally, our model also makes a qualitative prediction about the probability distribution of cable lengths at steady state. Namely, the feedback function near the steady state cable length, $L_{cable}^* = x^*L_{cell}$ can be Taylor expanded to

$$\frac{dL_{cable}}{dt} \approx f(x^*) + f'(x^*)\frac{L_{cable} - L_{cable}^*}{L_{cell}} = f'(x^*)\frac{L_{cable} - L_{cable}^*}{L_{cell}},\tag{8}$$

which shows that the strength of the feedback diminishes with cell length. This in turn implies that the steady state fluctuations of cable length will be larger in longer cells, which is consistent with data in *Figure 1G*. It is important to note that the above arguments pertain to cable length fluctuations over time, whereas the data in *Figure 1G* show cell-to-cell fluctuations in cable length, which could be influenced by cell-to-cell heterogeneity in some of the factors that affect cable assembly. Further experiments that carefully delineate between different sources of cable length fluctuations could provide more detailed tests of our model.

## Data and materials availability

Data are available in the main text or in the supplementary material. All images (*McInally et al., 2021b*) and source code (*McInally, 2021a*) are archived at Zenodo.

## Acknowledgements

We thank Ariel Amir, Lishibanya Mohapatra, James Moseley, Rob Phillips, Aldric Rosario, and Alison Wirshing for thoughtful discussions on cable length control and comments on the manuscript. We are also grateful to Brian Graziano for sharing the *cdc28-13^{ts}* strain. Funding: This material is based upon work supported by the the NSF Postdoctoral Research Fellowships in Biology Program under Grant No. 2010766 to SGM, by grants from NSF (DMR-1610737) and the Simons Foundation (http://

www.simonsfoundation.org/) to JK, a grant from the NIH to BLG (R35 GM134895), and the Brandeis NSF MRSEC, Bioinspired Soft Materials, DMR-2011486.

## Additional information

### Funding

| Funder | Grant reference number | Author |
|---|---|---|
| National Institutes of Health | R35 GM134895 | Bruce L Goode |
| National Science Foundation | 2010766 | Shane G McInally |
| National Science Foundation | DMR-1610737 | Jane Kondev |
| National Science Foundation | 2011486 | Jane Kondev<br>Bruce L Goode |
| Simons Foundation | | Jane Kondev |

The funders had no role in study design, data collection and interpretation, or the decision to submit the work for publication.

### Author contributions

Shane G McInally, Conceptualization, Data curation, Formal analysis, Funding acquisition, Investigation, Methodology, Writing - original draft, Writing - review and editing; Jane Kondev, Conceptualization, Supervision, Funding acquisition, Methodology, Writing - original draft, Project administration, Writing - review and editing; Bruce L Goode, Conceptualization, Supervision, Funding acquisition, Writing - original draft, Project administration, Writing - review and editing

### Author ORCIDs

Shane G McInally (iD) https://orcid.org/0000-0001-6145-4581
Jane Kondev (iD) https://orcid.org/0000-0001-7522-7144
Bruce L Goode (iD) https://orcid.org/0000-0002-6443-5893

### Decision letter and Author response

Decision letter https://doi.org/10.7554/eLife.68424.sa1
Author response https://doi.org/10.7554/eLife.68424.sa2

## Additional files

### Supplementary files

• Supplementary file 1. Yeast strains used in this study. The genotype, source, and related data are indicated for each strain used in this study.

• Transparent reporting form

### Data availability

All data points are shown in the main and supplemental figures, and all cell images and source code are archived at Zenodo.

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
