## [Decision Letter]

**Acceptance summary:**

This manuscript will be of interest to researchers in the fields of cell size control and the cytoskeleton. A combination of modelling and experimental data show that actin cables, which extend in budding yeast cells from the bud tip and bud neck, display an average length similar to the length of the cell, likely due to progressive decrease in their extension rate up to cell length. The distinct scaling relationship with cell length rather than volume may be a new paradigm that drives new investigations in related phenomena in other organisms.

**Decision letter after peer review:**

Thank you for submitting your article "Scaling of subcellular structures with cell length through decelerated growth" for consideration by *eLife*. Your article has been reviewed by 3 peer reviewers, and the evaluation has been overseen by a Reviewing Editor and Jonathan Cooper as the Senior Editor. The following individual involved in review of your submission has agreed to reveal their identity: Arjun Raj (Reviewer #1).

The reviewers have discussed their reviews with one another, and the Reviewing Editor has drafted this to help you prepare a revised submission. The vast majority of points raised can be addressed by rewriting and reanalyzing data already generated. However, a single new experiment has been sought by one of the referees, which I agree will further sharpen the models you are trying to advance.

Essential Revisions:

1. I'm not as familiar with the details of the models, but it seems that a key assumption is that f(L_cable_,L_cell_) = f(L_cable_/L_cell_), which allows for the cells to scale length appropriately. Can the authors speculate on potential mechanisms underlying this length scaling? If there is a gradient, what could form it with such a property?

2. The effect of Smy1 KO on initial elongation rate is modest. I think it would be useful to quantify the effect size and put it in the main text. Also, does the degree of increase quantitatively match the increased cable length observed? If not, can the authors speculate on the source of the discrepancy?

3. To uncouple cell length and cell volume, the authors could use mutants with abnormal cell shape (for instance using more ellipsoid or more rounded cells). Many such mutants have been described over the years in *S. cerevisiae*, so this should be a technically straightforward approach to take. This would allow to probe whether actin cable length indeed correlates better with cell length than the cubic root of volume (or other shape measurements).

4. An indication of how cables were selected for tracking and their number amongst how many cells would help better explain the extension rate measurements and increase confidence in the data. Extending this comparison to other mutants with altered cell shape as suggested above would also strengthen the conclusions.

5. Measurement of cable length and steady state length. The study depends heavily on the ability to measure actin cable length. But how cable length was measured in 3D is not presented with sufficient detail. I think it is essential to show convincing 3D images of cable end identification for the majority of actin cables in a cell. I am suggesting this since cables that reach the cell back may "turn around" and/or form bundles or unbundle through interactions with other cables.

6. The extension rate of actin cables that grow from the bud neck to the cell back is shown in movies and quantified in the figures. However, it's not clear if these cables eventually *stop* elongating. For example, in Figure 4D the extension rate does not decay to zero at the longest times. That seems to me to be an essential point since I would expect that if one is able to measure the length of most actin cables in the cell, then one should be able to see most of these cables stop extending.

7. I found the contrast between the two models, "boundary-sensing" and "balance-point" was not so sharp: the difference is on whether the extension rate decays more or less abruptly with distance along the cell. I also don't see if either of these models as presented is excluded by the data. The authors convincingly show that the cable extension rate decays with cable length, as would be expected from a balance-point model. However, as mentioned in comment 1, it is not clear where the extension rate decays to zero in graphs such as Figure 1F, 3B, 4D. If cable elongation ends abruptly at the longest time in these graphs, then this would indicate a boundary-sensing mechanism (so overall a combination/intermediate model).

8. I feel that the mechanism by which the actin cable system reaches a steady state needs to be clarified. We see a few cables that start from the bud neck and grow to the whole cell. Is this typical or are these rare cases? If rare, are they still representative? Do cables disappear to allow for new cable growth? Such processes change cable length so they are linked to the central theme of cable length regulation.

9. There is a lack of discussion and experimentation of known regulators of actin assembly and disassembly. Perhaps it's ok to leave the precise mechanism providing the feedback of cable deceleration for future work, however I feel that at least some discussion should be provided. For example, cofilin might sever filaments in age-dependent manner and cofilin-decorated filaments might be prone to breaking when buckled. Myosin motors are also known to regulate the extension rate of cables.

10. The authors don't measure cables shorter than 2 microns in Figure 1G, yet such cables should exist from the movies of cable growth versus time. I wonder if their measurement was biased towards the longest cables. In Figure 1D (cdc28-13 induced) there may be several short cables in addition to the long one. Perhaps a plot of total actin cable intensity vs distance would help.

*Reviewer #1:*

In this manuscript, the authors evaluate models for length determination of actin cables in yeast. The problem they pose is that actin cables must scale with the length of a cell, but given that length is a one dimensional measure, it cannot use "limiting pool" or "limiting factor" mechanisms in order to achieve such scaling. They first show that actin cables do indeed scale with cell length over a variety of cell cycle stages and perturbations. They then propose two models: a boundary sensing model and a balance-point model, the latter of which they show to be more consistent with the data. In line with their model, the rate of cable extension scaled with distance scaled by total cell length.

I found this manuscript to be a really nice example of quantitative cell biology. I think the problem of scaling of various cell components is fundamental and of great interest, and this work has framed an interesting example of the 1D scaling required for a linear component (actin cables). The paper is well written, the data is very solid with strong image analysis, and the claims are well supported.

*Reviewer #2:*

This is an interesting manuscript examining how the length of actin cables is set in budding yeast cells. The authors show that the average length of actin cables from bud neck to the distal pole of the mother cell is very similar to the length of the cell. The cable length appears to scale linearly with cell length in cells of diverse sizes (haploid, diploid, and cells artificially enlarged by a cell-cycle block), but cable length also scales with cell volume (with a power law – roughly with the cubic root of volume). The authors then propose a couple of possible models that would account for this scaling behaviour and show data that is in agreement with the idea that either assembly or disassembly rates are dependent on cable length, leading to a balance point at length equal that of the cell. The underlying molecular mechanism of this balance-point model is not examined, except for the finding that Smy1, which the authors previously proposed serves to limit cable length in an "antenna model" (Mohapatra et al., PLoS Comput Biol 2015), is not involved.

The correlation between cable length and cell length is very intriguing and suggestive that cell length is the relevant measure used by cells to set the length of their actin cytoskeleton. However, the presented data do not exclude that the relevant measure may be cell volume, with which actin cable length scales with cell volume using a more complex hypoallometric power law. To rigorously test whether cell length rather than cell volume is the relevant parameter, it would be necessary to decouple length from volume and test how actin cables length scales with these measurements.

The authors propose a balance-point model in which cable assembly and disassembly rates are balanced when the cable has the length of the cell due to either rate being cable length-dependent. They contrast this to a boundary model in which cable growth would abruptly stop when reaching the model cell distal pole. The experimental data to discriminate between these model focuses on measured cable extension rates in WT haploids, and normal-sized and enlarged cdc28-13 mutants. These measurements of actin cable extension by fluorescence microscopy are very challenging, and the authors should be commended for attempting it. However, they also raise uncertainty. Looking at the movies and the time-lapses, there are clearly many actin cables that do not behave as a simple birth at the neck followed by extension towards the opposite cell pole. Some of the tracking is also not certain – see for instance 17 to 18 s in movie 2: it looks like a big jump to a structure that was actually already present in the first of these two timepoints. This reduces somewhat the confidence in the conclusion that rate extension deceleration scales with cell length.

In summary, the finding of linear scaling between actin cable length and cell length is interesting, but whether cell length is the relevant parameter for this scaling is not fully established. There is a significant degree of uncertainty associated with the measurements of cable extension. While the current data is in agreement with a model in which cable extension diminishes with cable length, the mechanism of how this may be coupled to cell length is unknown.

*Reviewer #3:*

In this paper the authors find that the length of yeast actin cables scales with cell length. Cable length is studied using haploid, diploid, and cdc28-13 cells that grow abnormally large. Cable length extension rate is found to decrease with extension distance, at a different rate depending on cell size, supporting a "balance-point" mechanism of length regulation. The ratio between the shortest and longest cables in this work is a factor of order 2 (Figure 1F). This factor is significant to make this work interesting in the context of size regulation (though one can also be somewhat skeptical if a factor of 2 is large enough to unambiguously determine scaling mechanisms, given the multitude of actin regulators in cells that can provide such a change). Overall, this is a nicely written paper. However, I have several concerns as described below.

1) Measurement of cable length and steady state length. The study depends heavily on the ability to measure actin cable length. But how cable length was measured in 3D is not presented with sufficient detail. I think it is essential to show convincing 3D images of cable end identification for the majority of actin cables in a cell. I am suggesting this since cables that reach the cell back may "turn around" and/or form bundles or unbundle through interactions with other cables.

The extension rate of actin cables that grow from the bud neck to the cell back is shown in movies and quantified in the figures. However, it's not clear if these cables eventually *stop* elongating. For example, in Figure 4D the extension rate does not decay to zero at the longest times. That seems to me to be an essential point since I would expect that if one is able to measure the length of most actin cables in the cell, then one should be able to see most of these cables stop extending.

2) I found the contrast between the two models, "boundary-sensing" and "balance-point" was not so sharp: the difference is on whether the extension rate decays more or less abruptly with distance along the cell. I also don't see if either of these models as presented is excluded by the data. The authors convincingly show that the cable extension rate decays with cable length, as would be expected from a balance-point model. However, as mentioned in comment 1, it is not clear where the extension rate decays to zero in graphs such as Figure 1F, 3B, 4D. If cable elongation ends abruptly at the longest time in these graphs, then this would indicate a boundary-sensing mechanism (so overall a combination/intermediate model).

3) I feel that the mechanism by which the actin cable system reaches a steady state needs to be clarified. We see a few cables that start from the bud neck and grow to the whole cell. Is this typical or are these rare cases? If rare, are they still representative? Do cables disappear to allow for new cable growth? Such processes change cable length so they are linked to the central theme of cable length regulation.

4) There is a lack of discussion and experimentation of known regulators of actin assembly and disassembly. Perhaps it's ok to leave the precise mechanism providing the feedback of cable deceleration for future work, however I feel that at least some discussion would need to be provided. For example, cofilin might sever filaments in age-dependent manner and cofilin-decorated filaments might be prone to breaking when buckled. Myosin motors are also known to regulate the extension rate of cables.

---

## [Author Response]

Essential Revisions:1. I'm not as familiar with the details of the models, but it seems that a key assumption is that f(L_cable_,L_cell_) = f(L_cable_/L_cell_), which allows for the cells to scale length appropriately. Can the authors speculate on potential mechanisms underlying this length scaling? If there is a gradient, what could form it with such a property?

The manuscript now has been edited to include two potential molecular mechanisms leading to a scale-invariant gradient of actin depolymerizing activity (see end of main text).

In addition, how scale-invariant gradients are potentially established is discussed in more depth in a new theory manuscript on BioRxiv (see: https://doi.org/10.1101/2021.05.18.444733). This theory paper proposes that regulatory proteins diffuse in the cytoplasm and are captured and transported to the cell pole by motors moving along the cell cortex (a form of the ‘antenna’ mechanism – see minor comment #4 below), producing a scale-invariant gradient of these regulatory proteins. The simple idea is that upon release at the pole, proteins diffuse a typical distance λ≈Dτ , until they are captured by the motor proteins at the cell surface; here D is the protein diffusion constant in the cytoplasm and τ is the time until a protein is captured at the surface by a motor protein. In a spherical cell of radius Lcell,τ≈Lcell2/D, which leads to λ≈Dτ ≈Lcell. In other words, the combined action of cytoplasmic diffusion and motor transport at the cell surface, leads to a spatial protein gradient with a decay length to λ≈Lcell. Assuming this is a gradient of depolymerizing activity, and further assuming that the cable assembly rate is length-independent, leads to the scaling of the cable extension rate with the assumed scaling property. While no such gradient of cytoskeletal depolymerizing activity has been described in yeast thus far, its existence and role in regulating lengths of flagella in *Giardia* have been demonstrated (see: 10.7554/eLife.48694).

2. The effect of Smy1 KO on initial elongation rate is modest. I think it would be useful to quantify the effect size and put it in the main text. Also, does the degree of increase quantitatively match the increased cable length observed? If not, can the authors speculate on the source of the discrepancy?

The manuscript has been edited to include the direct comparison of the effect size for the increased cable length and cable extension rates measured in *smy1Δ* cells. As indicated in the edited text, the effect size of these phenotypes is similar, and thus we conclude that the observed increase in cable length is likely due to the initial increase in cable extension rate. Additionally, this comment inspired us to move the experimental test of our quantitative model (using *smy1∆*) from the supplement to the main text; these data previously appeared in the supplemental mathematics section, but we realized that they could be easily overlooked in that location. While the Smy1 data do not identify the molecular mechanism controlling cable deceleration, they confirm that increased initial cable extension rate will lead to a proportional increase in the initial deceleration of the cable, a specific prediction of our model in Equation 7. We believe that moving this section to the main text will strengthen our paper and be helpful to readers in light of the reviewer’s comment.

3. To uncouple cell length and cell volume, the authors could use mutants with abnormal cell shape (for instance using more ellipsoid or more rounded cells). Many such mutants have been described over the years in S. cerevisiae, so this should be a technically straightforward approach to take. This would allow to probe whether actin cable length indeed correlates better with cell length than the cubic root of volume (or other shape measurements).

To address this, we have performed an analysis of cable lengths in cells of different shape, as indicated by their different aspect ratios (ratio of cell length to width). This analysis allows us to compare cables in cells of the same length but with different volumes. These data (see new ‘Figure 1 —figure supplement 2’) show that cable length is not affected by differences in cell shape, supporting our conclusion that cable length scales with cell length, rather than other dimensions such as surface area or cell volume.

4. An indication of how cables were selected for tracking and their number amongst how many cells would help better explain the extension rate measurements and increase confidence in the data. Extending this comparison to other mutants with altered cell shape as suggested above would also strengthen the conclusions.

We have revised the methods to provide additional information regarding the inclusion criteria for cables used in live cell imaging analyses. Briefly, cables were only included in our analysis if we could observe their extending tips emerge from the bud neck, and track their tips until they stopped growing. The number of cables analyzed, and the number of independent experiments is indicated in each figure legend. The typical number of cables tracked in an individual cell (by live imaging) is ~1-2 due to technical constraints, such as the relatively short time (~2 minutes) that we can acquire images before the signal begins to bleach, and before there is substantial drift in the z-dimension that prevents accurate cable tip tracking. With future methodological improvements, it may become possible to track larger numbers of cables per cell, and for longer periods of time.

5. Measurement of cable length and steady state length. The study depends heavily on the ability to measure actin cable length. But how cable length was measured in 3D is not presented with sufficient detail. I think it is essential to show convincing 3D images of cable end identification for the majority of actin cables in a cell. I am suggesting this since cables that reach the cell back may "turn around" and/or form bundles or unbundle through interactions with other cables.

We revised the methods to provide additional information about how cable lengths were measured. We clarify that cables were not measured in 3D, but that the 3D stack was used to reliably determine where cables emerged from the bud neck and where their tips were in the mother cell. In addition, we have included annotated maximum intensity projects (see ‘Figure 1 —figure supplement 1A’) that display all of the cables traced in example cells. Additional annotated cell images are available in our imaging dataset uploaded to Zenodo (10.5281/zenodo.4791679).

To address the possibility that cables ‘turn around’, we have computed cable end-to-end distance and tortuosity. The end-to-end distance of a cable is the distance between the distal tip of the cable and its starting point back at the bud neck. Tortuosity is the ratio of cable length (obtained from the trace) to its end-to-end distance, where a perfectly straight cable has a value of one. This analysis shows that the majority of cables do not turn around or make sharp turns, and have a mean cable tortuosity of 1.2-1.3 (see ‘Figure 1 —figure supplement 1’). In our live imaging experiments, we have never observed individual cables interacting and dynamically bundling or unbundling with each other, although our data do not exclude the possibility that this could occur.

6. The extension rate of actin cables that grow from the bud neck to the cell back is shown in movies and quantified in the figures. However, it's not clear if these cables eventually *stop* elongating. For example, in Figure 4D the extension rate does not decay to zero at the longest times. That seems to me to be an essential point since I would expect that if one is able to measure the length of most actin cables in the cell, then one should be able to see most of these cables stop extending.

When we “measure the length of most actin cables in the cell” we are using super-resolution imaging of chemically fixed cells. In contrast, when we perform live imaging of cables we are only able track a subset of the cables in a cell for technical reasons stated earlier (see comment #4; also see comment #8 below). That said, we expect that the reason cable extension rates (by live imaging) do not decay to zero is a consequence of tracking the tips of cables, which can ‘wiggle’ or diffuse laterally while their length does not change. Additionally, all of the cable deceleration profiles we have measured stop at ~0.1um/sec, which is close to the size of a single pixel (0.133um/pixel) in our videos. Thus, we expect that this represents the limit of detection in our assays and should be interpreted as cables no longer extending.

7. I found the contrast between the two models, "boundary-sensing" and "balance-point" was not so sharp: the difference is on whether the extension rate decays more or less abruptly with distance along the cell. I also don't see if either of these models as presented is excluded by the data. The authors convincingly show that the cable extension rate decays with cable length, as would be expected from a balance-point model. However, as mentioned in comment 1, it is not clear where the extension rate decays to zero in graphs such as Figure 1F, 3B, 4D. If cable elongation ends abruptly at the longest time in these graphs, then this would indicate a boundary-sensing mechanism (so overall a combination/intermediate model).

As stated above, we do not think that the final cable extension rate falls to zero (see response to comment #6). Further, cable tips were not observed to abruptly stop. With these clarifications in mind, we believe that the difference between the ‘boundary-sensing’ and ‘balance-point’ models is fairly sharp. In one model, cables exhibit a constant rate of extension until they physically encounter the cell boundary and then they halt. In the other model, cable extension rate is constantly changing, starting out fast as the cable emerges from the bud neck and then slowing as a function of cell length – until the cable length matches the length of the cell (see ‘Figure 2C and 2D’).

8. I feel that the mechanism by which the actin cable system reaches a steady state needs to be clarified. We see a few cables that start from the bud neck and grow to the whole cell. Is this typical or are these rare cases? If rare, are they still representative?

As mentioned above, due to technical limitations we are only able to track a subset of the cables in a cell by live imaging (see response to comment #6). However, given our analysis of cables in fixed cells, where the majority of cables grow to reach the back of the cell, we are fairly confident that our live tracking of cables extending to reach the rear of the cell are representative of typical cable behavior.

Do cables disappear to allow for new cable growth? Such processes change cable length so they are linked to the central theme of cable length regulation.

How cables disappear to allow for new cable growth is an important question that we hope to address in the future. Little is known about how cables are disassembled in vivo once they reach the back of the cell. To date, these questions have been difficult to approach experimentally, in part due to the same technical limitations mentioned above (e.g., our imaging is limited to ~2 minutes before photobleaching and drift prevent accurate image acquisition). We anticipate that improvements in fluorescent tags (to mark cables) and better control of the microscope stage will allow for longer analyses in the future, and make it possible to begin answering these types of questions.

9. There is a lack of discussion and experimentation of known regulators of actin assembly and disassembly. Perhaps it's ok to leave the precise mechanism providing the feedback of cable deceleration for future work, however I feel that at least some discussion should be provided. For example, cofilin might sever filaments in age-dependent manner and cofilin-decorated filaments might be prone to breaking when buckled. Myosin motors are also known to regulate the extension rate of cables.

We have revised our Discussion to include several possible molecular mechanisms controlling cable deceleration (see response to comment #1). This includes the possibility of an activity gradient of cofilin and/or other actin disassembly factors, where their activity is higher at the rear of the cell. We also acknowledge that actin filament aging could play a role in the observed scaling relationship (cable length with cell length). However, this would require filament aging (in cables) to be cell length dependent, given that initial cable extension rates are similar in cells of different length. As such, cells would need to have a mechanism for slowing π release (from the F-actin in cables) in longer cells, or alternatively, accelerating π release from cables in shorter cells.

10. The authors don't measure cables shorter than 2 microns in Figure 1G, yet such cables should exist from the movies of cable growth versus time. I wonder if their measurement was biased towards the longest cables. In Figure 1D (cdc28-13 induced) there may be several short cables in addition to the long one. Perhaps a plot of total actin cable intensity vs distance would help.

We fully expect that there are cables smaller than 2 µm, but they are also growing much faster than longer cables, and are therefore less frequently observed. Additionally, as touched upon above (see response to comment #8) we don’t know what fraction of all the cables in a cell are undergoing de novo assembly at any given time.